# Necessity of Uncertainty Quantification for Audio driven healthcare diagnosis

**Shubham  Kulkarni, Hideaki Watanabe, Fuminori Homma**
Sec. 2, AI Application Development
Sony Group Corporation
Tokyo, Japan 141-8610
`shubham.a.kulkarni@sony.com`

## Abstract

Deep learning excels in analyzing multi-modal signals for healthcare diagnostics but lacks the ability to quantify confidence in the predictions, which can lead to overconfident, erroneous diagnoses. In this work, we propose to predict model output independently and estimate the corresponding uncertainty. We present a unified audio-driven disease detection framework incorporating uncertainty quantification (UQ). This is achieved using a Dirichlet density approximation for model prediction and independent kernel distance learning in feature latent space for UQ. This approach requires minimum modifications to existing audio encoder architectures and is extremely parameter efficient compared to k-ensemble models. The uncertainty-aware model improves prediction reliability by producing confidence scores that closely match the accuracy values. Evaluations using the largest publicly available respiratory disease datasets demonstrate the advantage of the proposed framework in accuracy, training and inference time over ensemble and dropout methods. The proposed model improves speech and audio analysis for medical diagnosis by identifying and calibrating uncertainties, enabling better decision-making and risk assessment. This is shown by high uncertainty scores at low model accuracy.

## 1   Introduction

The increase in general awareness and interest in speech technologies for disease diagnosis has generated significant growth in recorded public health datasets Song et al. (2023); Novikova and Balagopalan ([n. d.]) across different modalities such as audio, imaging and time series (EEG). As the healthcare industry increasingly embraces data-driven approaches, the accurate interpretation of these subtle and complex multi-modal signals has become paramount for informed decision-making and improved patient outcomes. However, for these models to be useful in practical implementation, the outputs of such models must be explainable for medical decision making Miller (2019). Multi-modal medical datasets have been extensively researched for the task of disease diagnosis, symptom identification and monitoring Kulkarni et al. (2023); Wang and Wang (2022); Bae et al. (2023). Popularly, large-scale convolutional neural network (CNN) architectures Demir et al. (2020) such as ResNet Gairola et al. (2021); Bengs et al. ([n. d.]) trained on spectrogram images of audio inputs are used for this task. Recently, direct waveform speech encoders (Wav2Vec Baevski et al. (2020), and PASE Ravanelli et al. (2020)) have shown improved speech feature representations for respiratory monitoring Kulkarni et al. (2023). After featurisation, a classification layer followed by softmax is used to produce output scores. However, fixed softmax scores may result in fundamentally incorrect outputs without indicating that the estimate is uncertain. Thus, achieving a statistically nuanced

Workshop on Bayesian Decision-making and Uncertainty, 38th Conference on Neural Information Processing Systems (NeurIPS 2024).

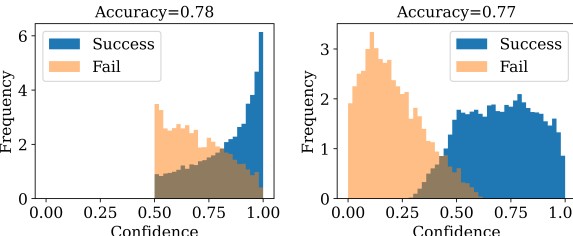

Figure 1: Calibration histograms for speech-driven COVID classifier (left) and uncertainty aware model (right) coloured according to prediction accuracy

understanding of model outputs via uncertainty quantification (UQ) is crucial in safety-critical applications such as disease detection.

This can be illustrated with a simple example of speech-driven disease detection. Input audio can be either "healthy" or COVID-"positive." A softmax-based classifier gives scores that express the likelihood of two different classes. Figure 1 (left) shows a histogram of the softmax scores coloured according to the correctness of the predicted output. The plot shows that irrespective of correctness of the prediction, the output confidence is always greater than 50%. The confidence score for two inputs (one predicted correctly and another incorrectly) lying on a vertical will be exactly same (`healthy = 0.89, positive = 0.11`) and (`healthy = 0.11, positive = 0.89`). Without UQ model, there is no way to decide the reliability of either prediction based on just softmax probabilities. An independent UQ estimate can quantify high uncertainty for false predictions, as shown in Figure 1 (right). An uncertainty-aware audio classification model enables 1) prediction of confidence scores independent of model outputs and 2) calibration of model such that estimated uncertainty closely follows model accuracy.

In this work, we present a novel framework for uncertainty-aware disease detection using speech and non-speech inputs through quantification and disentanglement of sample uncertainty and model calibration. The framework comprises of a probabilistic classification head on top of a self-supervised audio encoder and model uncertainties are quantified using a feature distance-based metric. A training scheme is proposed to optimize uncertainty estimation independent of model prediction or classification training. A novel formulation of learnable transformation matrix in latent space is used to maximise feature space diversity for distance calculation. Evaluations show that the uncertainty-aware model produces low confidence scores at low accuracy values, thus improving output reliability. Experiments on the largest public respiratory disease datasets show that the proposed UQ model is generalizable, computationally efficient at training and enables fast evaluation during inference without sacrificing classification performance. Specifically, our contributions are as follows -

- Advocate the use of a probabilistic classifier in place of softmax scores to quantify irreducible uncertainties inherent in learning problem for audio-driven disease diagnosis and medical decision making
- Emphasize the necessity of model calibration for reducible uncertainties in audio-driven disease diagnosis. we show that combining probabilistic classifier simple k-ensembles (even with small k=5) significantly improves model calibration score
- Propose a novel single inference method of uncertainty quantification with minimal changes to large encoder models for high-fidelity datasets such as audio and speech. The proposed model performs as well as k-ensembles at a fraction of compute and memory costs

To best of our knowledge, this is the first systematic study of uncertainties quantification and model calibration associated with audio driven disease diagnosis.

## 2   Model

Lets denote $a(t) \in \mathcal{A}$ as an input audio waveform and $(y_j = y_j + \epsilon_j)$ is its corresponding noisy label which takes a value from label space $j \in \{1, \ldots \mathbf{J}\}$ and $\epsilon_j$ is the label noise due to data gathering process or the noise inherent to the mapping problem $G : \mathcal{A} \to \mathbf{J}$. We decompose above function mapping as $G = h \circ f$, where, $f : \mathcal{A} \to \mathbf{R}^n$ indicates a deep audio feature encoder. The feature

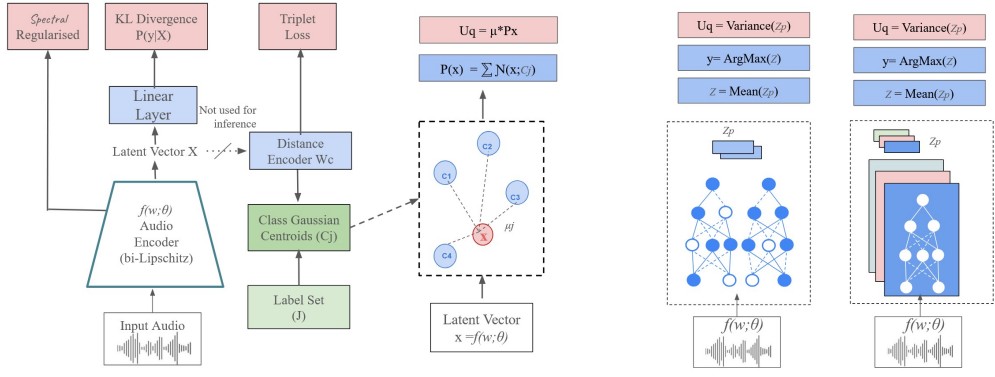

((a)) Single-shot UQ framework;
label prediction (left) and uncertainty calibration (right)

((b)) Multi-shot UQ models;
MC Dropout (left) and Ensemble (right)

Figure 2: Proposed framework for uncertainty quantification (UQ) of audio driven disease detection

encoder gives embedding vectors $X_w(a) \in \mathcal{R}^d$. The uncertainty aware classification head $h : X \rightarrow y$ gives a prediction over class labels $P[y|x] = h(X)$.

The proposed uncertainty quantification (UQ) framework, illustrated in Figure 2, consists of two parts:

1. A probabilistic classifier $h$ trained to output concentration parameters of Dirichlet distribution over the softmax layers. This classifier head is used on top of a regularised deep audio feature extractor ($f$), which produces latent embedding $X$.

2. An uncertainty aware calibration training to estimate UQ as a function of feature space density. We use a novel learnable Mahalanobis distance-based metric, which ensures the latent space is bi-Lipschitz continuous and captures a measure of data distribution.

In the subsequent sections, we describe these two component of the proposed UQ framework

## 2.1 Probabilistic Classifier

A deterministic softmax classifier only outputs a single scaled vector $s(x)$ corresponding a input $x$ such that $\sum_j s(x) = 1$. In contrast, the probabilistic classifier head is trained to predict a vector of concentration parameters $\alpha = (\alpha_1 \dots \alpha_J)$ one for each class label $j \in J$, and a strength parameter $\alpha_0 := \sum_j (\alpha_j)$. This set of concentration parameters define a Dirichlet distribution $\text{Dir}(\alpha)$ with probability density given by equation 1, where $\Gamma(\cdot)$ denotes *Gamma* function.

This is used to sample a class probability vector $\boldsymbol{p}$ as a random vector $\boldsymbol{p} \sim \text{Dir}(\boldsymbol{\alpha})$, At the inference time, a sample from Dirichlet distribution gives indicative probability $p_j$ of input $x$ belonging to class $j$. The expected probability (mean) and the variance for a single input x is given by

$$\text{Dir}(\boldsymbol{p}|\boldsymbol{\alpha}) = \frac{\Gamma(\alpha_0)}{\prod_{c=1}^{C} \Gamma(\alpha_c)} \prod_{c=1}^{C} p_c^{\alpha_c - 1} \quad \leftrightarrow \quad \begin{aligned} \mu(x) &:= \mathbf{E}[p_j] = \frac{\alpha_j}{\alpha_0} \\ \sigma^2(x) &:= \frac{\alpha_j(\alpha_0 - \alpha_j)}{\alpha_0(\alpha_0 + 1)} \end{aligned} \quad (1)$$

Thus, the classifier head is a model with uncertainty that outputs two quantities corresponding to label distribution, the mean $\mu(x)$ and the variance $\sigma(x)$. The sampling based output stems from key insight that softmax based classifier cannot capture output categorical probability but a distribution over categorical softmax (i.e. Dirichlet) can be used to formulate deep learning as evidence acquisition problem Sensoy et al. (2018); DeVries and Taylor (2018).

The classification head is trained using unweighted combination of negative log likelihood term $\mathcal{L}^{NLL}$ and a KL-divergence term, following the Sensoy et al. (2018); Bachstein et al. (2019). Appendix covers Loss function derivations and final expressions. Upon training the classifier model using above loss, we obtain predictive distribution parameters - mean $\mu(x)$ and variance $\sigma(x)$. However

this quantity only gives the output label probability of a given input for a fixed model. Considering the original function mapping problem $G : \mathcal{A} \to \mathbf{J}$ and the decomposition $G = h \circ f$, the probabilistic classifier $h$ is a single sample from a possibly large intractable hypothesis space $\mathcal{H}$. Further, the audio encoder $f$ is parametrised by a set weights $\mathbf{W}$. In the supervised setting a point estimate of vector $W$ is obtained by empirical risk maximisation of an objective function. In Bayesian modelling Lakshminarayanan et al. (2016); Gal and Uk (2016), uncertainties in this point estimate, are computed by assuming that the weights $w$ follow a prior distribution $Pr(w)$. Subsequently, the model training process leads to posterior distribution $P(w|D)$. The trained model $f_w(x)$ uses this posterior distribution to calculate the estimated output $y$. The measure of uncertainty, UQ, is given by the expected value and variance of the prediction $f_w(x)$ over the posterior density distribution of $w$. However, for high-dimensional datasets such as audio and speech, accurate modelling of the density function $P(w|D)$ is impossible, given the complexity and non-linear nature of weights $w$ of audio classification models Hernández-Lobato and Adams (2015).

Figure 2(b) shows two approaches for approximating the intractable posterior density $P(w|D)$ by introducing diversity into model evaluations. The feature encoder generates a fixed and deterministic encoding vector $X$ for the input audio signal. Model uncertainty is quantified by analysing the variance of the outputs obtained through multiple forward passes of diverse models. In **Monte Carlo (MC) dropout** Gal and Ghahramani (2016); Xiao and Wang (2019), probabilistic ($p = 0.1$) dropout layers between non-linear layers of the network are activated during inference resulting in variable outputs. Whereas, in **Deep ensemble** Lakshminarayanan et al. (2016), k-different models ($k = 5, 15$) are trained using different subsets of the dataset. The ensemble prediction is the average soft-max outputs from the individual models.

Combining classification head with multi-forward pass inferences in equation 1, we get a series of means $\mu_k(x)$ and variances $\sigma_k^2(x)$, where $k \in [1, K]$ are number of different ensembles or inferences of Figure 2(b). These samples are combined to form a single predictive uncertainty estimate $Var[X]$ for input $X$ as an empirical expectation over all inferences $k$. A combination of Deep Ensemble

$$
\begin{aligned}
\text{Var} * (\mathbf{x}) = & \; \tfrac{1}{K} \sum_k \sigma_k^2(\mathbf{x}) & + & \; \tfrac{1}{K} \sum_k \mu_k^2(\mathbf{x}) - \mu_*^2(\mathbf{x}) \\
= & \; \mathbf{E}_k[\sigma_k^2(\mathbf{x})] & + & \; \mathbf{E}_k[\mu_k^2(\mathbf{x})] - \mathbf{E}_k[\mu_k(\mathbf{x})]^2 \\
= & \; \underbrace{\mathbf{E}_k[\sigma_k^2(\mathbf{x})]}_{\text{Aleatoric Uncertainty}} & + & \; \underbrace{\text{Var}_k[\mu_k(\mathbf{x})]}_{\text{Epistemic Uncertainty}}
\end{aligned}
$$

and Dirichlet Probabilistic classifier gives an estimate for the Irreducible Aleatoric Uncertainty and Model Uncertainty (Epistemic). However, it is neither possible to treat each term separately nor to reduce epistemic part of uncertainty. Despite the limitations, the k-ensemble approach is shown to be state-of-the art for uncertainty prediction on several benchmarks Mukhoti et al. (2021). Both these methods improve performance and uncertainty estimation through model diversity but incur high computational costs during training and inference. In next section, we describe the second part of the proposed framework - an alternative to k-ensemble for quantifying approximate Epistemic uncertainty in single forward pass.

## 2.2 Single Inference Uncertainty Quantification

In contrast to multiple feed-forward evaluation models, we propose single-shot UQ estimation using latent feature maps produced by the encoder as a representation of the class conditional distribution. A distance measure in the feature space of the model has shown to be useful for the detection of out-of-distribution examples Venkataramanan et al. (2023) and uncertainty estimation Lee et al. ([n. d.]); van Amersfoort et al. (2020). However, these methods suffer from three key problems namely feature collapse van Amersfoort et al. (2020), class imbalance Venkataramanan et al. (2023), smoothness and sensitivity Lee et al. ([n. d.]). We first describe the proposed single shot approach with intuitive modifications to training scheme that address the aforementioned problems.

The uncertainty estimation flow is shown in Figure 2(a). A centroid vector $Z \in \mathcal{R}^m$ is initialised randomly and assigned to each label class in a set of classes $J$. Let $X_t(i) \in \mathcal{R}^d$ be the set of audio encodings of a mini-batch during training. A distance transformation matrix $W_j(m, d)$ is initialised using a Gaussian prior per class, where $d$ is the feature encoding dimension and $m < d$ is the size of the centroid vector. Weight matrix $W_j$ acts as a learnable linear dimensionality reduction

on feature vectors, enabling a compact representation for distance computation Ren et al. (2021); Venkataramanan et al. (2023). The class-dependent nature of $W_j$ enables class separation in latent space and is crucial for minimising the likelihood of feature collapse.

A weighted feature distance $D_j$ between the model output and centroids is computed as:

$$D_j(X_t, Z_j) = \sqrt{\frac{||W_j X_t - Z_j||^2}{2m\sigma_j^2}}$$

where length scale $\sigma_j$ is a trainable parameter and acts as class dependent normalising hyper-parameter.

If the matrix $W$ is assumed to be Identity Matrix the above formulation computes Mahalanobis distance (MD) from the centroids. The learnable nature of $W$ acts as an adaptive dimensionality reduction on the latent space $X$ and the output $WX$ can be expected to represent global distributions as well as class dependent local distributions.

During the forward pass, a class label for each sample is given by softmax of distance scores $y_i = \mathbf{Argmin} Z_j(X_i)$ as the maximum correlation (minimum distance) between data point $X_i$ and class centroids $Z_j$. For the UQ estimate, the set of Mahalanobis distances is normalised through the division of maximum class distance. The model uncertainty is given by mixture of the Gaussian models fitted at each class centroid $d_{\mathbf{UQ}} = \sum j \mathcal{N}(D_j | z_j, \sigma_j)$.

The class centroids, $Z_j$, are updated for every mini-batch of training using an exponential moving average of the feature vectors of data points corresponding to class $j$:

$$Z_{t+1,j} = \gamma Z_{t,j} + \frac{1}{n_j}(1-\gamma)\sum_i (W_j X_i)$$

where $n_j$ is number of samples in the $j^{th}$ class, and $\gamma$ is a hyper-parameter similar to momentum gradient descent. After each update, the class vectors are normalised such that $||Z_j||_2 = 1$.

Class dependent **triplet Loss** formulation is used to maximise the distance between distinct class centroids and minimise intra-class separation, following Kumar et al. (2020); Hermans et al. (2017) . Audio embeddings obtained from the encoder network were used as an anchor point $X_a$. Let $Z_a$ be the centroid vector of the class corresponding to true label $y_a$, while $Z_j$ indicates remaining centroid vectors such that $\{j \in J \forall j \neq a\}$, The loss with margin $\epsilon \in (0.1 - 0.5)$ is given by

$$\mathcal{L}_{triplet} = \sum_{a,j} \mathbf{max}\left(||WX_a - Z_a|| - ||WX_a - Z_j|| + \epsilon, 0\right)$$

During the training process, this loss is averaged over a mini-batch of data points, the class centroids are updated to new locations as per predicted labels and stochastic gradient descent (SGD) is performed for $\theta$ and $W_j$. Audio encoder output latent vectors usually have high dimensions and the above loss may suffer poorly due to involved distance computation. Low rank nature of $W$ ensures that distance computation in above loss function is sensible.

**Feature Regularisation** High dimensional feature space embedding suffer from feature collapse and feature redundancy in latent space which can adversely affect uncertainty prediction Liu et al. (2020); van Amersfoort et al. (2020). These problems can be alleviated by encouraging latent space smoothness and sensitivity, or alternatively by regularising the the weights $W$ to follow bi-Lipschitz condition Liu et al. (2020)

$$L_1 * ||x_1 - x_2||_X \leq ||f_W(x_1) - f_W(x_2)||_H \leq L_2 * ||x_1 - x_2||_X$$

This ensures the mapping $||f_W(x_1) - f_W(x_2)||_H$ has meaningful correspondence in input space with respect to a well defined distance measure $||x_1 - x_2||_X$ Liu et al. (2020). This condition also ensures smoothness in latent space such that the audio embeddings are not too sensitive to small variations in input.

We use spectral normalisation to enforce bi-Lipschitz condition during UQ training, following the analysis Smith et al. (2021); Liu et al. (2020)that adding spectral normalisation before each convolution layer leads to bi-Lipschitz condition. Apart from being simpler in implementation (with minor changes to encoder architecture such as replacing L2 norm layer by spectral norm), spectral normalisation is significantly faster Smith et al. (2021) and is more stable during training compared to Jacobian Gradient penalty implemented in van Amersfoort et al. (2020).

# 3 Experiments

We will now demonstrate the utility of proposed framework in quantifying uncertainties of audio driven disease diagnosis. We first start with a brief description of datasets, evaluation criterion and implementation details. (detailed description and data histograms are covered in Appendix)

## 3.1 Datasets

We conduct extensive experiments using two popular audio-driven healthcare diagnosis datasets.

The **ICBHI** Rocha et al. (2018) dataset is the largest publicly available respiratory audio repository recorded from 128 patients with a total of 6898 labelled breathing cycles (Label distribution 3642 normal, 1864 crackle, 886 wheeze, and 506 cycles as both). The highly unbalanced dataset constitutes a 4-class audio classification task.

**COSWARA** Sharma et al. (2020) consists of a diverse set of manually curated audio records from 2635 individuals, of which 1819 are SARS-CoV-2 negative, 674 are positive subjects, and the remaining unlabelled or noisy samples are filtered out. Speech recordings of numbers (1-20) counted at a fast pace were used for this 2-class classification and disease detection task.

## 3.2 Self-supervised Audio Encoder

Self-supervised learning (SSL) is an attractive approach for healthcare audio datasets where the data size is limited and manual annotation is expensive Sharma et al. (2020); Rocha et al. (2018). Three different SSL models are employed as audio encoders for the empirical evaluation. First, an image-based **ResNet-50** is used as the backbone with a residual block of two $3 \times 3$ convolution layers and a skip connection between each block. The network is trained on the self-supervised task of spectral feature prediction and reconstruction of the log-Mel spectrogram. Further, **Wav2Vec** Baevski et al. (2020) and **PASE** Ravanelli et al. (2020) are used as direct waveform feature encoders. Each encoder is pre-trained on the respective SSL pretext task and used to obtain latent representations from raw audio.

Let $a(t) \in \mathcal{A}$ be an input audio waveform and $y = 1, \ldots J$ be its corresponding label. The feature encoder gives embedding vectors $X_w(a) \in \mathcal{R}^d$, where $d = 256$ is the fixed latent dimension.

## 3.3 Preprocessing

All audio files were resampled to a fixed rate of 22.05kHz. The ICBHI respiratory sounds were cropped/padded to max a length of 7s Gairola et al. (2021); Kulkarni et al. (2023), while COSWARA speech were fixed to 10s length Sharma et al. (2020). In the case of ResNet, each audio was transformed to log Mel-spectrogram using 128 frequency bins. An input size of (128, 350) was used for ICBHI, whereas, for COSWARA, the input size was (128, 500). For both cases, the dataset was divided into three non-overlapping portions such that the test set (20%) and validation set (20%) contained audio records from different patients than that of the train set (60%).

## 3.4 Evaluation

For measuring accuracy of model, **sensitivity** ($\frac{TP}{TP+FN}$), and **specificity** ($\frac{FP}{FP+TN}$) scores were used. Each score measures class-wise prediction accuracy in the case of the unbalanced dataset. The notations $TN, FN$ denote true and false negative rates and $TP, FP$ denote true and false positive rates, respectively. Average of these two scores ($\frac{SP+SN}{2}$) was used for comparison with SoTA models Rocha et al. (2018). The area under the receiver operating curve (**AUROC**) was used as an indicative probability of correctly classifying a randomly selected unseen sample.

Most common measure predictive uncertainty is Expected Calibration error (**ECE**). Low ECE indicates model accuracy closely follows predicted uncertainty estimates, i.e. low model accuracy in high-uncertainty regions and vice versa. To calculate ECE on a test set, all test samples are grouped in $k = 10$ equal bins according to uncertainty scores. ECE was calculated as the absolute sum of differences between expected model confidence and accuracy for each bin. A small ECE indicates better performance as the model accurately quantifies uncertainties in its prediction. Experiments

show that ECE values drastically reduce with the proposed UQ implementation while maintaining the model's accuracy.

## 4    Results and Discussion

The first goal of the experiments is to answer the question 'whether model uncertainty score follows model accuracy'. Figure 7 show reliability diagrams of ICBHI 4-class classification model using PASE encoder as backbone. The model output is divided in equally spaced bins according to estimated confidence score for each bin. Reliability plots show the average accuracy of the examples in each corresponding confidence bin. We also visualise the confidence scores (1- uncertainty) with class conditional histograms of correctly and incorrectly classified outputs. The proposed model reliably predicts high uncertainty misclassified examples while producing high uncertainty for accurately classified examples.

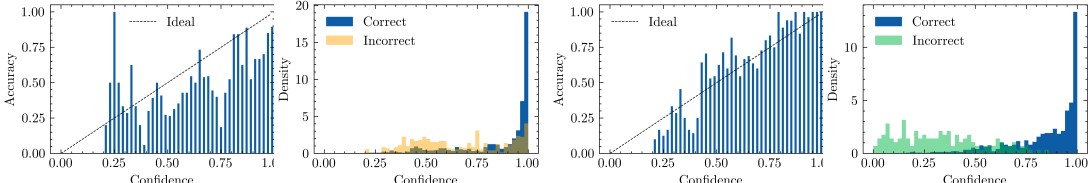

Figure 3: Reliability diagrams before and after feature distance based uncertainty calibration. Plots show that proposed models predicts UQ scores that closely follow the model accuracy. (low confidence scores for low accuracy data regions and vice versa)

Table 1: Performance comparison of different base encoder models with and without uncertainty estimation (ICBHI)

| Model | Base (Dirichlet) | | Base+UQ | |
|---|---|---|---|---|
| | AUROC | ECE | AUROC | ECE |
| PASE | $0.835_{\pm 0.01}$ | $0.121_{\pm 0.01}$ | $0.905_{\pm 0.01}$ | $0.055_{\pm 0.01}$ |
| Wav2Vec | $0.778_{\pm 0.02}$ | $0.148_{\pm 0.01}$ | $0.812_{\pm 0.02}$ | $0.069_{\pm 0.01}$ |
| ResNet | $0.746_{\pm 0.01}$ | $0.106_{\pm 0.01}$ | $0.862_{\pm 0.01}$ | $0.041_{\pm 0.01}$ |

A similar analysis is conducted for different choices of base feature encoder (Table 1) by considering the ECE (error) and AUROC (accuracy) of ICBHI respiratory classification task using different audio encoders (ResNet, PASERavanelli et al. (2020) and Wav2Vec Baevski et al. (2020)) with and without UQ estimation. A significant reduction in ECE values is observed among all three feature models. This means the model is more uncertain for false predictions and more confident for correct outputs. ResNet achieves higher relative improvement compared to direct waveform-based audio encoders. This is due to ResNet having higher embedding dimension compared to SSL encoders and thus adversely affecting the class conditional density estimation in latent space Ren et al. (2021). The low rank class-wise linear transformation enables distribution aware low dimensional transformation, improving both AUROC and ECE score.

**Classification Accuracy** and dataset variability of uncertainty aware models are compared in Table 2. Bootstrapping is used to compute the maximum confidence interval. The proposed model shows a significant advantage in ECE prediction over other UQ methods with marginal improvements in model accuracy.

**An ablation study** was conducted to study the incremental effects of various loss functions by fixing the feature encoder of the proposed UQ model. Table 3a displays the ECE and accuracy improvements with each additional loss term. A significant reduction in ECE error is observed upon the inclusion of triplet loss term for both datasets.

**Compute efficiency** of the proposed method, in terms of the number of parameters (in Millions) and inference time (in milliseconds), is compared with those of popular UQ models in Table 3b. The scores show the expected inference time for a single sample averaged over the test set compared

Table 2: Evaluation of the UQ framework for two different datasets with fixed feature encoder (PASE)

| Model | ECE | SN(%) | SP(%) | AUROC |
|---|---|---|---|---|
| **ICBHI$_{\text{4- class}}$** | | | | |
| Base | $0.161_{\pm0.01}$ | $79.8_{\pm4.71}$ | $50.5_{\pm6.21}$ | $0.782_{\pm0.01}$ |
| MC Drop. | $0.064_{\pm0.01}$ | $79.6_{\pm5.31}$ | $42.6_{\pm5.91}$ | $0.732_{\pm0.02}$ |
| Ensemble | $0.051_{\pm0.01}$ | $83.1_{\pm3.71}$ | $57.7_{\pm1.91}$ | $0.888_{\pm0.01}$ |
| **Our (UQ)** | $\mathbf{0.045}_{\pm0.01}$ | $\mathbf{82.1}_{\pm4.07}$ | $\mathbf{55.1}_{\pm3.75}$ | $\mathbf{0.823}_{\pm0.01}$ |
| **COSWARA$_{\text{2-class}}$** | | | | |
| Base | $0.191_{\pm0.02}$ | $96_{\pm3.32}$ | $72.9_{\pm2.21}$ | $0.781_{\pm0.01}$ |
| MC Drop. | $0.074_{\pm0.01}$ | $96_{\pm5.59}$ | $70_{\pm4.19}$ | $0.951_{\pm0.01}$ |
| Ensemble | $0.060_{\pm0.01}$ | $96.6_{\pm3.15}$ | $77.9_{\pm4.98}$ | $0.964_{\pm0.01}$ |
| **Our (UQ)** | $\mathbf{0.058}_{\pm0.01}$ | $\mathbf{95.9}_{\pm4.81}$ | $\mathbf{74.6}_{\pm2.91}$ | $\mathbf{0.961}_{\pm0.01}$ |

Table 3: Ablation study (a) of proposed UQ framework to study effects of modification terms, along with network size (Millions) and inference time (sec) of different UQ models(Results on non-intersecting splits of ICBHI dataset with PASE as feature encoder)

(a) Ablation study

| **Model** | **ECE** | **AUROC** |
|---|---|---|
| Old (Softmax) | $0.158_{\pm0.01}$ | $0.741_{\pm0.02}$ |
| Base (Dirichlet) | $0.149_{\pm0.01}$ | $0.876_{\pm0.01}$ |
| + KL Divergence | $0.104_{\pm0.01}$ | $0.921_{\pm0.02}$ |
| + Triplet loss | $0.086_{\pm0.01}$ | $0.923_{\pm0.02}$ |
| + Regularisation | $0.065_{\pm0.01}$ | $0.918_{\pm0.01}$ |

(b) Network size

| **Method** | **AUROC** | **Params** | **Inference** |
|---|---|---|---|
| Base (logits) | 0.782 | 26M | 1.8 ms |
| MC Dropout | 0.732 | 26M | 4.3 ms |
| Ensemble - 5 | 0.888 | 132M | 9.8 ms |
| Ensemble - 15 | 0.891 | 395M | 29 ms |
| Mahalanobis | 0.823 | 26M | 2.1 ms |

against AUROC scores. In this case, PASE is used as the base model. The ensemble model performed well but was extremely slow at inference time with a large number of parameters, increasing the storage and compute overhead. The Mahalanobis distance-based uncertainty estimation enables lightweight and fast inference while improving model accuracy.

Table 4: Comparison with SoTA models and recent studies on four-class respiratory anomaly detection (ICBHI dataset)

| Method | Performance | | |
|---|---|---|---|
| | **SN(%)** | **SP(%)** | **Acc.** |
| ResNet Gairola et al. (2021) | 40.1 | 72.3 | 56.2 |
| ResNeST Wang and Wang (2022) | 70.4 | 40.2 | 55.3 |
| CNN8-Pt Ren et al. (2022) | 72.9 | 27.8 | 50.4 |
| ResNet Chang et al. (2022) | 69.9 | 35.8 | 52.9 |
| CVAE-Tr Bae et al. (2023) | 81.7 | 43.1 | 62.4 |
| Our (UQ) **ECE- 0.058** | $\mathbf{82.1}_{\pm4.07}$ | $\mathbf{55.1}_{\pm3.75}$ | $\mathbf{68.5}_{\pm3.92}$ |

**Comparison** with state-of-the-art (SoTA) models for ICBHI 4+class respiratory sound classification task is presented in Table 4. The proposed model improved the accuracy scores over the current SoTA by 6.1%. A validation set sensitivity score of 82.1% indicates the ability to correctly identify true positives from unseen patient samples recorded using different digital stethoscopes. Accounting for the uncertainties not only provides a nuanced understanding of output but also improves model performances for audio-driven disease diagnosis.

## 5   Uncertainty Visualisation and Decision Making

The outputs produced by sampling from Dirichlet distribution (output of probabilistic classifier for a single input) satisfy the property that $\sum_j (p_j) = 1$, where $p_j$ is probability $P[y = j|X]$. For a three class problem (ICBHI - wheeze, crackle, healthy) , each of these samples fall on the 2D plane defined by $\sum_j (p_j) = 1$. Figure 4 shows uncertainty visualisations on the simplex plane. This uncertainty

is sample specific (data/ aleatoric uncertainty) indicating inherent label noise or ambiguity in the samples.

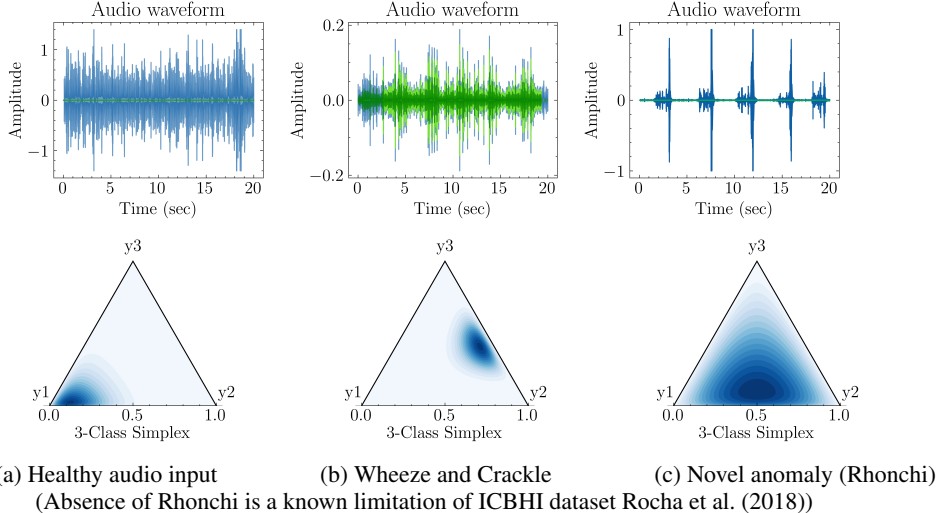

(a) Healthy audio input      (b) Wheeze and Crackle      (c) Novel anomaly (Rhonchi)
(Absence of Rhonchi is a known limitation of ICBHI dataset Rocha et al. (2018))

Figure 4: Plots visualising data uncertainty corresponding to each input audio sample. The network predicts Dirichlet distribution parameters ($\alpha$) in single forward pass which are then used to plot probability density over the simplex.

At the same time, the model uncertainty (predictive) is given by mixture of the Gaussian models fitted at each class centroid $d_{\mathbf{UQ}} = \sum j \mathcal{N}(D_j | C_j, \sigma_j)$. This estimate is independent model prediction at a given sample. This is a measure of learning capacity of model for current input and can also be used as OOD indicator (epistemic uncertainty). A threshold on the UQ score can be used as a decision factor for audio-driven medical diagnosis. If the predicted UQ value is higher than this threshold, the model is not sufficiently confident in its prediction; thus, the disease diagnosis output is rejected. In such cases, second or multiple evaluations using re-recording of input audio samples are recommended. If the resulting uncertainty, after multiple empirical evaluations, is still higher than the threshold, then the particular sample is selected for clinical or manual diagnosis. This avoids the risk of erroneous predictions via uncertainty quantification. As a result, the proposed framework improves the performance of audio-driven disease detection system along with patient safety. (Such threshold based rejection was not used during experiments and results, however it can be a useful tool for medical decision making)

## 6   Conclusion

In this work, a framework for uncertainty-aware disease diagnosis was proposed using speech and non-speech inputs. The UQ framework enables confidence scoring to improve the reliability of model outputs.Evaluations of the popular COSWARA and ICBHI datasets illustrate the superiority of the proposed model over the popular ensemble and Monte Carlo dropout method. Using the same ResNet backbone, the UQ aware model outperformed softmax-based SoTA models for respiratory diseaseBae et al. (2023) without using data driven oversampling techniques. Using the UQ model for the ICBHI dataset, an improvement of 6.1% was observed over the SoTA models. Furthermore, for speech-driven COVID detection, quantifying data uncertainty improves AUROC scores by 18.1%. The UQ model performs well on unseen datasets, as seen from results on non-intersecting inter-patient data splits, and is equally applicable to more general datasets. Results also show the effectiveness and applicability of the Mahalanobis distance-based metric for different general-purpose audio encoders. Finally, the proposed framework enables fast and lightweight UQ estimation, making it more suitable for implementation in mobile and IoT devices for continuous health monitoring owing to its small size and lower number of trainable parameters.

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

# Appendix / supplemental material

## A   Algorithm

We consider the probabilistic function learning problem between input audio space $X \in \mathcal{X}$ and corresponding discrete label space $Y \in \mathcal{Y}$, where $X$ and $Y$ are random variables. We denote $x \in \mathcal{X}$ and $y \in \mathcal{Y}$ as data samples from joint space $\mu_{XY} = (X, Y)$, with a joint distribution function denoted by $\mu_{XY}$. A model trained with uncertainty aware classification tries to approximate the conditional probability distribution $\mu_{Y|X} = \mathbf{P}[Y = y|X]$.

The training dataset $D_n = \{(x_i, y_i) \forall i = 1 \ldots N\}$ (a subset of joint space $\mu_{XY}$) is used to train an estimator $\hat{Y} = g(X; W)$, where $g$ denotes a neural network with parameter $W$. Further, the collected dataset itself can be inherently noisy or error-prone, which results in data uncertainty (also known as Aleatoric uncertainty). The noise in the dataset is indicated by $x_i = \hat{x} + \eta_i$ and $y_i = \hat{y} + \epsilon_j$ where the the observed noisy dataset is given by, $D_n = \{(x_i, y_i)\}$. This aleatoric uncertainty is irreducible and can only be estimated as expected variance in the output for a fixed input $X$, and a given estimator $f(x|W)$. However is not the only source of uncertainty in the estimator, the output variance does not capture the uncertainty in estimator or the learning process itself.

The total predictive probability, can be expanded as follows -

$$\mu_{Y|X} = \mathbf{P}[Y = y|X] \tag{2}$$
$$= \mathbf{P}[Y = y|X, D_n]P[X \in D_n] \tag{3}$$
$$.. + \mathbf{P}[Y = y|X, D_n]P[X \notin D_n] \tag{4}$$
$$\tag{5}$$

The second term in above equation signifies distribution uncertainty, i.e. uncertainty associated with limitations of training data. This can be reduced by obtaining more training data i.e. by minimising $P[X \notin D_n]$. Assuming the dataset $D$ is used to learn a function $y = f(x; W)$, parametrized by the weights $W$, the first term can further be expanded as follows -

$$\mu_{Y|X,D} = \mathbf{P}[Y = y|X, D_n] \tag{6}$$
$$= \int P(y|X, w)dP(w, D) \tag{7}$$
$$= \int P(y|X, w)P(w|D)dw \tag{8}$$
$$\tag{9}$$

This integral is called as inference using posterior density $P[w|D]$, this computation involves test time optimisation, by formulating closed form of posterior density. The second term, posterior in above equation can be decomposed as

$$P(w|D) = \frac{\mathbf{P}[D|W]P[W]}{P[W]} \tag{10}$$
$$P(D) = \int P(D|w)dP(w) \tag{11}$$
$$= \int P(D|w)P(w)dw \tag{12}$$
$$\tag{13}$$

This integral is called as marginal integration to compute a form for posterior density from a presumed prior $p[w]$. Often this marginal is intractable for most non trivial forms of likelihood functions $p[D|W]$. The inference integral is often approximated using multiple forward pass via Dropout or Ensemble modelling. The goal of the proposed distance based model is to provide an efficient single forward pass alternative to approximate the marginal and inference integrals.

Given that we can view an ensemble member as a single deterministic model and vice versa, this provides an intuitive explanation for why single deterministic models report inconsistent and widely varying predictive entropies and confidence scores for OoD samples for which a Deep Ensemble would report high epistemic uncertainty (expected information gain) and high predictive entropy.

Assuming that $p(y|x,\omega)$ only depends on $p(y|x)$ and $\mathbb{I}[Y;w|x]$, we model the distribution of $p(y|x,\omega)$ (as a function of $\omega$) using a Dirichlet distribution $Dir(\alpha)$ which satisfies:

$$p(y|x) = \frac{\alpha_i}{\alpha_0} \tag{14}$$

$$H[Y|x] - \mathbb{I}[Y;w|x] = \psi(\alpha_0 + 1) \tag{15}$$

$$\tag{16}$$

Then, we can model the softmax distribution using a random variable $\mathbf{p} \sim Dir(\alpha)$ as:

$$P(y|x;w) \stackrel{\approx}{\sim} Cat(\mathbf{p}). \tag{17}$$

The variance $VarH[Y|x;w]$ of the softmax entropy for different samples $x$ given $p(y|x)$ and $\mathbb{I}[Y;w|x]$ is then approximated by $VarY|\mathbf{p}$: This is the estimate of Aleatoric Uncertainty in the model. For the random variable, $\mathbf{p} \sim \mathrm{Dir}(\alpha)$, the expected entropy $\mathbb{E}_{\mathbf{p}\sim\mathrm{Dir}(\alpha)}\mathbb{H}_{Y\sim\mathrm{Cat}(\mathbf{p})}[Y]$ of the categorical distribution $Y \sim \mathrm{Cat}(\mathbf{p})$ is given by

$$\mathbb{E}_{\mathrm{p}(\mathbf{p}|\alpha)}\mathbb{H}[Y \mid \mathbf{p}] = \psi\left(\alpha_0 + 1\right) - \sum_{y=1}^{K} \frac{\alpha_i}{\alpha_0}\psi\left(\alpha_i + 1\right)$$

Proof. Applying the sum rule of expectations and 3 from 1.1 we can write

$$\mathbb{E}\mathbb{H}[Y \mid \mathbf{p}] = \mathbb{E}\left[-\sum_{i=1}^{K}\mathbf{p}_i\log\mathbf{p}_i\right] = -\sum_{i}\mathbb{E}\left[\mathbf{p}_i\log\mathbf{p}_i\right]$$
$$= -\sum_{i}\frac{\alpha_i}{\alpha_0}\left(\psi\left(\alpha_i + 1i - \psi\left(\alpha_0 + 1\right)\right)\right.$$

The result follows after rearranging and making use of $\sum_i \frac{\alpha_i}{\alpha_0} = 1$.

# B    Base Model

## B.1    Feature Encoder

A feature encoder serves as base model (backbone) of the framework. The feature encoder serves as an indicative audio classification backbone. The proposed framework can accommodate any state of the art audio encoder and does not require any modification in the feature encoder training process and architecture. Specifically, when $a(t) \in \mathcal{A}$ be an input audio waveform, the feature encoder gives embedding vectors $X_w(a) \in \mathcal{R}^d$, where $d = 256$ is the fixed latent dimension. Next we explain the base encoders used for experiments

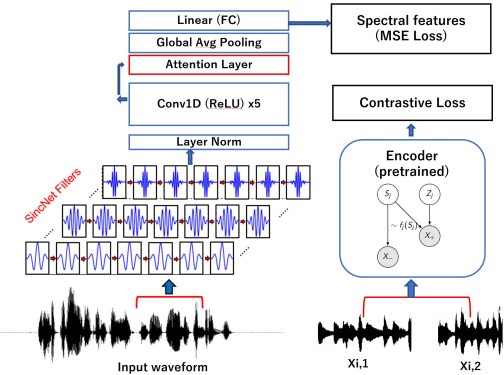

Figure 5: Self supervised feature encoder architecture for PASE+ Ravanelli et al. (2020)

## B.2  Wav2vec2.0

wav2vec is a self-supervised learning model trained to learn representations of raw audio waveforms directly, without relying on manual transcriptions or labels. Wav2vec employs contrastive learning to learn powerful representations from raw audio inputs. A more recent version, wav2vec 2.0 introduces a more sophisticated approach by masking portions of the latent space rather than the raw audio. Wav2Vec 2.0 significantly improves upon the quality of learned representations and demonstrates exceptional performance in downstream speech tasks. We use Wav2vec2.0 as one of the backbone feature extractor in the proposed framework. The Wav2vec model predicts the masked latent representations, encouraging it to capture rich contextual information. The output embedding dimension of the Wev2vec encoder is fixed to be 256.

## B.3  PASE +

Problem Agnostic Speech Encoder (PASE) is another self supervised audio feature encoder which employs multiple neural networks, termed "workers," to tackle various self-supervised tasks. These workers contribute to learning rich and discriminative representations. To ensure robust feature vectors with respect to small variations in input audio, PASE+ introduces an online speech distortion module that artificially corrupts the input audio, forcing the encoder to learn more invariant features. As shown in Figure 5 PASE+ also uses bidirectional attention layers to combine convolution ouputs to better capture both short-term and long-term speech dynamics.

## B.4  ResNet

An image-based **ResNet-50** is used as the backbone with a residual block of two $3 \times 3$ convolution layers and a skip connection between each block. The network is trained on the self-supervised task of spectral feature prediction and reconstruction of the log-Mel spectrogram. The network consists of a series of convolution layers. Each of these layers is defined with 64 channels, kernel strides (5, 2, 2, 2, 2, 2, 2), and kernel widths (7, 3, 3, 3, 3, 3, 2, 2), respectively, followed by batch normalization and ReLU activation. The interval between two sequential samples in the feature encoder output Z is 15ms, and the receptive audio field is 20 ms. The output from convolution layers is concatenated and passed to a multi-head attention layer and a fully connected layer with an embedding size of 256. Like PASE Ravanelli et al. (2020), the final linear layer is used to predict speech features such as log power spectrum (LPS), MFCCs, prosody, 40 FBANKS and 40 Gammatone features. The architecture is pre-trained on an open source audio dataset called Audioset **?**, consisting of a wide variety of input sounds ranging such as birds, coughs, speech and machine sounds. During pretraining, the model predicts a set of 12 supervised tasks consisting of regression and binary feature banks such as log power spectrum (LPS), MFCCs, prosody and Gammatone features. This pretraining ensures that the ResNet representations are tuned capture short and long-range audio dynamics over a wide variety of input sounds. These representations are proven to outperform spectrogram-based large CNN models and standard acoustic features for different classification and speech recognition tasks Ravanelli et al. (2020). These representations are then frozen to compute encoding for respiratory cycle datasets. Experiments show that no significant improvement are observed with additional complete finetuning on the ICBHI dataset during the training phase compared to the frozen representation.

# C  Criteria

## C.1  Probabilistic Classifier

We train the classification using unweighted combination of negative log likelihood term $\mathcal{L}^{NLL}$ and a KL-divergence term, following the Sensoy et al. (2018); Bachstein et al. (2019). Appendix covers Loss function derivations and final expressions.

The loss function expressions of $\mathcal{L}^{NLL}$ and $\mathcal{L}^{KL}$ are respectively

$$\mathcal{L}^{NLL} = \sum_{c=1}^{C} y_c (\log(\alpha_0) - \log(\alpha_c)) \tag{18}$$

$$\mathcal{L}^{KL} = \log\left( \frac{\Gamma(\sum_{c=1}^{C} \tilde{\alpha}_c)}{\Gamma(C) \prod_{c=1}^{C} \Gamma(\tilde{\alpha}_c)} \right)$$
$$+ \sum_{c=1}^{C} (\tilde{\alpha}_c - 1) \left( \psi(\tilde{\alpha}_c) - \psi\left( \sum_{c=1}^{C} \tilde{\alpha}_c \right) \right) \tag{19}$$

in which $\tilde{\alpha}_c = y_c + (1 - y_c)\alpha_c$ and $\psi(\cdot)$ is *Digamma* function.

These two losses can viewed intuitively as a union of **Bayes Risk Approximation** losses, which is defined with respect to class conditional density prediction. We use Bayes risk formulation from PAC learning nomenclature as given below,

$$\mathcal{L}_i(\Theta) = \sum_{j=1}^{K} (y_{ij} - \mathbf{E}\,[p_{ij}])^2 + \mathrm{Var}\,(p_{ij}) \tag{20}$$

$$= \sum_{j=1}^{K} \underbrace{(y_{ij} - \alpha_{ij}/S_i)^2}_{\mathcal{L}_{ij}^{\mathrm{err}}} + \underbrace{\frac{\alpha_{ij}\,(S_i - \alpha_{ij})}{S_i^2\,(S_i + 1)}}_{\mathcal{L}_{ij}^{\mathrm{Var}}} \tag{21}$$

$$= \sum_{j=1}^{K} (y_{ij} - \hat{p}_{ij})^2 + \frac{\hat{p}_{ij}\,(1 - \hat{p}_{ij})}{(S_i + 1)}. \tag{22}$$

## C.2 Uncertainty Calibration Network

During the forward pass, a class label for each sample is given by softmax of distance scores $y_i = \mathbf{Argmin} Z_j(X_i)$ as the maximum correlation (minimum distance) between data point $X_i$ and class centroids $Z_j$. For the UQ estimate, the set of Mahalanobis distances is normalised through the division of maximum class distance. The model uncertainty is given by mixture of the Gaussian models fitted at each class centroid $d_{\mathbf{UQ}} = \sum j \mathcal{N}(D_j | z_j, \sigma_j)$.

The class centroids, $Z_j$, are updated for every mini-batch of training using an exponential moving average of the feature vectors of data points corresponding to class $j$:

$$Z_{t+1,j} = \gamma Z_{t,j} + \frac{1}{n_j}(1 - \gamma) \sum_i (W_j X_i)$$

where $n_j$ is number of samples in the $j^{th}$ class, and $\gamma$ is a hyper-parameter similar to momentum gradient descent. After each update, the class vectors are normalised such that $||Z_j||_2 = 1$.

Class dependent **triplet Loss** formulation is used to maximise the distance between distinct class centroids and minimise intra-class separation, following Kumar et al. (2020); Hermans et al. (2017) . Audio embeddings obtained from the encoder network were used as an anchor point $X_a$. Let $Z_a$ be the centroid vector of the class corresponding to true label $y_a$, while $Z_j$ indicates remaining centroid vectors such that $\{j \in J \forall j \neq a\}$, The loss with margin $\epsilon \in (0.1 - 0.5)$ is given by

$$\mathcal{L}_{triplet} = \sum_{a,j} \mathbf{max}\,(||W X_a - Z_a|| - ||W X_a - Z_j|| + \epsilon, 0)$$

During the training process, this loss is averaged over a mini-batch of data points, the class centroids are updated to new locations as per predicted labels and stochastic gradient descent (SGD) is performed for $\theta$ and $W_j$.

## C.3 Evaluation

For measuring accuracy of model, **sensitivity** ($\frac{TP}{TP+FN}$), and **specificity** ($\frac{FP}{FP+TN}$) scores were used. Each score measures class-wise prediction accuracy in the case of the unbalanced dataset. The notations $TN, FN$ denote true and false negative rates and $TP, FP$ denote true and false positive rates, respectively. Average of these two scores ($\frac{SP+SN}{2}$) was used for comparison with SoTA models Rocha et al. (2018). The area under the receiver operating curve (**AUROC**) was used as an indicative probability of correctly classifying a randomly selected unseen sample.

Most common measure predictive uncertainty is Expected Calibration error (**ECE**). Low ECE indicates model accuracy closely follows predicted uncertainty estimates, i.e. low model accuracy in high-uncertainty regions and vice versa. At high thresholds, the model is tolerant of low confidence predictions, and thus, the model accuracy should decrease. At low uncertainty thresholds, the model should have high accuracy and confidence scores. To calculate ECE on a test set, all test samples are grouped in $k = 10$ equal bins according to uncertainty scores. ECE was calculated as the absolute sum of differences between expected model confidence and accuracy for each bin. A small ECE indicates better performance as the model accurately quantifies uncertainties in its prediction. Experiments show that ECE values drastically reduce with the proposed UQ implementation while maintaining the model's accuracy. The expected difference between

$$ECE = \sum_{k=1}^{10} \frac{n_k}{n} |ascore(B_k) - uscore(B_k)|$$

where $n_k$ is number of samples in $k^{th}$ bin, $ascore$ and $uscore$ are average accuracy and uncertainty estimates for each bin $B_k$. Experiments show that calibration error drastically reduces with the addition of UQ models while maintaining the model accuracy in AUROC scores. It can be interpreted as the probability that a positive example (in-distribution) will have a higher detection score than a negative example (out-of-distribution).

## D Datasets

We conduct extensive experiments using two popular audio-driven healthcare diagnosis datasets.

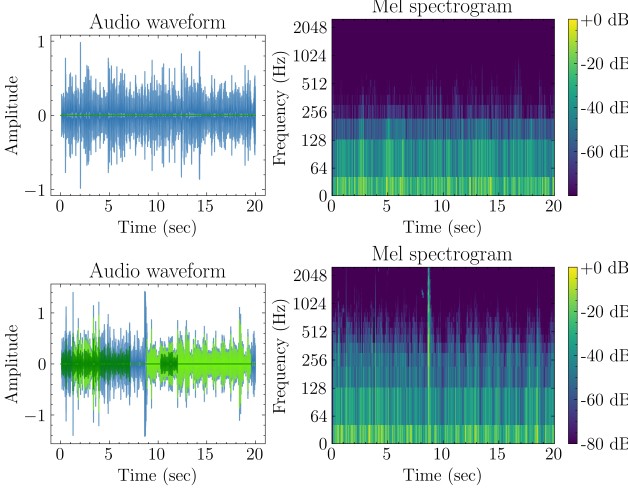

Figure 6: Audio samples showing varying degrees of anomalous (green) and healthy (blue) classes illustrating the necessity of uncertainty quantification

The **ICBHI** Rocha et al. (2018) dataset is the largest publicly available respiratory audio repository recorded from 128 patients with a total of 6898 labelled breathing cycles (Label distribution 3642 normal, 1864 crackle, 886 wheeze, and 506 cycles as both). The highly unbalanced dataset constitutes a 4-class audio classification task. Figure 6 shows audio samples showing varying degrees of anomalous (orange) and healthy (blue) classes. The input sample contains illustrating the necessity of uncertainty quantification. We share training and validation sets of this dataset for SoTA comparison.

**COSWARA** Sharma et al. (2020) consists of a diverse set of manually curated audio records from 2635 individuals, of which 1819 are SARS-CoV-2 negative, 674 are positive subjects, and the remaining unlabelled or noisy samples are filtered out. Speech recordings of numbers (1-20) counted at a fast pace were used for this 2-class classification and disease detection task. The dataset is manually curated and has approximately 10% noisy audio samples.

All audio files were resampled to a fixed rate of 22.05kHz. The ICBHI respiratory sounds were cropped/padded to max a length of 7s Gairola et al. (2021); Kulkarni et al. (2023), while COSWARA speech were fixed to 10s length Sharma et al. (2020). In the case of ResNet, each audio was transformed to log Mel-spectrogram using 128 frequency bins. An input size of (128, 350) was used for ICBHI, whereas, for COSWARA, the input size was (128, 500). For both cases, the dataset was divided into three non-overlapping portions such that the test set (20%) and validation set (20%) contained audio records from different patients than that of the train set (60%). [1]

## E  E Experiments

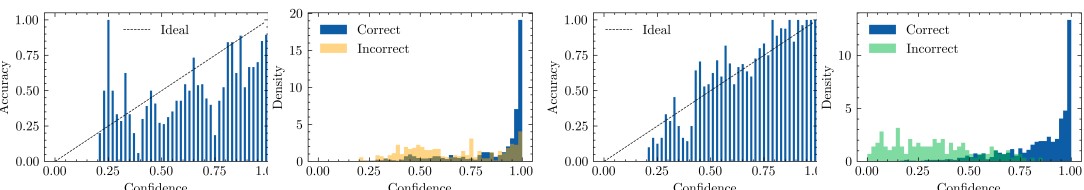

Figure 7: Reliability diagrams before and after feature distance based uncertainty calibration. Plots show that proposed models predicts UQ scores that closely follow the model accuracy. (low confidence scores for low accuracy data regions and vice versa)

The proposed framework is trained independently in two stages. The distance transformation matrix $W$ and audio feature encoders were optimised during the first stage of training process. It is important to note that the goal of feature encoder training not to represent state-of-the-art for any particular task – the goal is to demonstrate value of quantifying model uncertainty independent of the model prediction. We will show that across various of of-the-shelf audio feature encoders, the addition of UQ framework enables significant gains in model utility by not only quantifying model confidence but also reducing the calibration error of the model. This point is reinforced here using 2D synthetic dataset. In second stage of training the probabilistic classifier is optimised using KL divergence loss. In this second stage we show that, using off-the-shelf encoders it is possible to achieve and state of the art performance on popular disease diagnosis task. Classwise confusion matrix show the effectiveness of the probabilistic classifier for class imbalanced classification scenario.

### E.1  UQ on 2D dataset

In the proposed uncertainty quantification framework weighted feature distance $D_j$ between the model output and centroids is computed as:

$$D_j(X_t, Z_j) = \sqrt{\frac{||W_j X_t - Z_j||^2}{2m\sigma_j^2}}$$

where length scale $\sigma_j$ is a trainable parameter and acts as class dependent normalising hyper-parameter.

If the matrix $W$ is assumed to be Identity Matrix the above formulation computes Mahalanobis distance (MD) from the centroids. The learnable nature of $W$ acts as an adaptive dimensionality reduction on the latent space $X$ and the output $WX$ can be expected to represent global distributions as well as class dependent local distributions.

Figure 9 shows comparison of uncertainty estimates obtained using distance based metric and ensemble based model. In contrast to multiple feed forward evaluation models, a single shot

---

[1]Training and validation set labels shared as supplementary material.

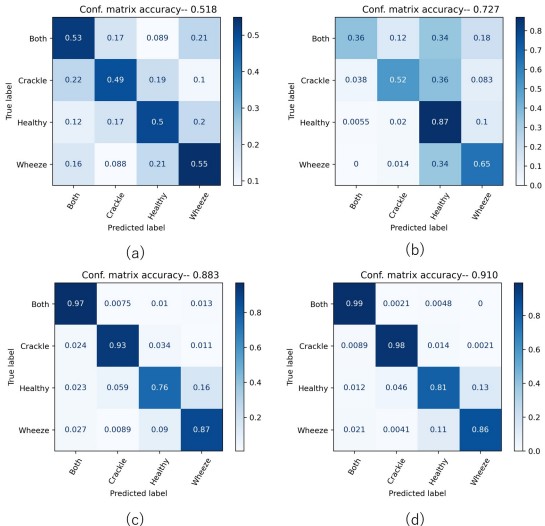

Figure 8: Evaluation of individual anomalous class performance

estimation of distance function in feature space gives an approximation of class conditional density. A Mahalanobis distance metric Lee et al. ([n. d.]); Venkataramanan et al. (2023) between output and the class centroids is has already been shown to act as an approximation of class conditional density and outperform empirical ensemble models for the task of OOD detection.

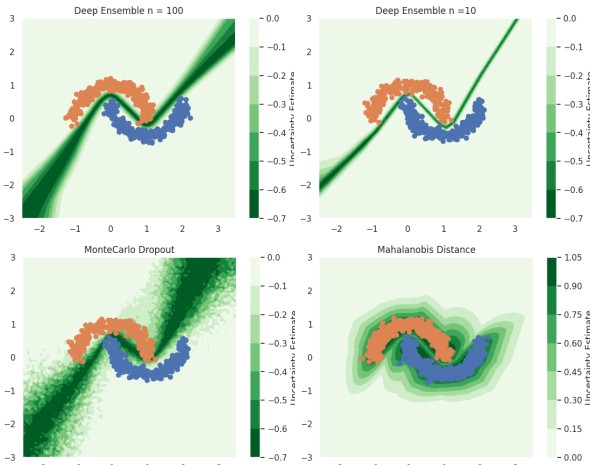

Figure 9: Comparison of proposed UQ model with popular Bayesian methods using confidence heat maps (Green) for a 2D synthetic dataset

