# OpenReview forum: "Uncertainty Quantification and Calibration for Audio-driven Disease Diagnosis"
_NeurIPS.cc/2024/Workshop/BDU — NeurIPS BDU Workshop 2024 Poster_

### Official Review · Reviewer_wgb9 · 2024-09-26
**Great Work (Length Concern (9 Pages vs. 4-Page Abstract)**

**Rating:** 9
**Confidence:** 3

**Review:**

**Major Concern (Note to AC/PC):** Paper consists of 9pgs of content instead of 4-page extended abstract.

### Pros
- This paper is one of the first work addressing UQ in audio classification.
- It proposes a novel framework for UQ in audio-driven disease detection, it uses a Probabilistic Classifier and a Single Inference method to measure uncertainty which utilizes latent feature maps created by the encoder to depict the class-conditional distribution.
- The authors validate their methodology on two medical audio detection datasets: ICBHI(4 class) and COSWARA(2 class) demonstrating a significant decrease in calibration error with their proposed technique.
- The proposed framework achieves SOTA performance on the ICBHI4 dataset.

### Cons
- **Major:** The primary section spans 9 pages, while the appendix, references, and NeurIPS checklist extend the total to 26 pages.
- The writing can be somewhat difficult to follow due to various formatting issues.(Especially citations with lack of appropriate spacing)

### Summary
This paper could serve as a valuable contribution to the workshop audience, provided it can be condensed to 4 pages.

---

### Official Review · Reviewer_7hHA · 2024-10-03
**Uncertainty Quantification and Calibration for Audio-driven Disease Diagnosis**

**Rating:** 6
**Confidence:** 5

**Review:**

This paper proposes a novel framework for incorporating uncertainty quantification (UQ) into audio-driven disease diagnosis models.

Pros:

Addresses a crucial limitation: Deep learning models often lack the ability to quantify confidence in predictions, leading to overconfident and erroneous diagnoses. This work proposes a method to address this by estimating model and data uncertainties.
Unified framework: The proposed approach integrates UQ with audio-driven disease detection using a Dirichlet density approximation and independent kernel distance learning.
Improved model reliability: The uncertainty-aware model produces confidence scores that closely match accuracy values, increasing overall model reliability.
Minimal modifications: The proposed method requires minimal modifications to existing audio encoder architectures and is computationally efficient compared to ensemble models.
Generalizability: Evaluations on large public respiratory disease datasets demonstrate the framework's generalizability and efficiency.

Cons:

Limited experimental details: While the paper mentions using three self-supervised audio encoders (ResNet-50, Wav2Vec, PASE), it lacks specifics on hyperparameter tuning, training details, and validation strategies.
Comparison with existing UQ methods: The paper mentions limitations of existing UQ methods like dropout and ensemble models but doesn't explicitly compare the proposed method's performance with these approaches.

Originality:

The combination of a probabilistic classifier with a learnable distance metric for UQ in audio-driven disease diagnosis appears to be novel.
The focus on single-shot UQ estimation with minimal alterations to existing encoder architectures offers a potentially efficient approach.


Significance:

This work contributes to the development of more reliable and interpretable AI models for medical diagnosis.
By enabling quantification of uncertainties, the proposed framework can improve decision-making in healthcare settings by highlighting areas of low model confidence.
The emphasis on computational efficiency further enhances the framework's potential for real-world applications.

---

### Decision · Program_Chairs · 2024-10-09

Accept (Poster)